# REDUCING DISTORTIONS IN REAL WORLD IMAGE SUPER RESOLUTION USING ATTENTION

## ABSTRACT

In the real-world super-resolution (SR), we focus on enhancing perceptual quality beyond conventional resolution enhancement, aiming to address real-world degradations using Generative Adversarial Networks (GANs). GANs hold the potential for restoring fine details in low-resolution images, yet their generative nature may introduce distortions. Our contribution involves a novel approach that improves perceptual quality while minimizing distortions in real-world SR images through the strategic use of residual connections and an attention map. This approach has a simple structure and can be used to improve the performance of previously published SR models by using them as a backbone. We show in our experiments that our proposed method successfully reduces the distortions derived from GANs, thus improving the perceptual quality.

## 1 INTRODUCTION

Image super-resolution aims to elevate a lower-resolution image to a higher-resolution counterpart while retaining the essence of the original image (Dong et al., 2014). However, contemporary approaches, exemplified by classical SR methods, often produce high-resolution images with deficiencies stemming from their inherent limitations.

One of the key drawbacks of classical SR models is the emergence of artifacts, such as the notorious ringing phenomenon, when confronted with input images significantly deviating from the domain characteristics of the training dataset (Chen et al., 2018). Furthermore, the reliance on the Peak Signal-to-Noise Ratio (PSNR) as a primary loss metric during classical SR model training proves inadequate in accurately gauging the degree of high-resolution image quality (Blau & Michaeli, 2018). This deficiency becomes evident when applying classical SR methods to real-world images replete with diverse degradations, manifesting unforeseen artifacts. Consequently, classical SR models struggle with instability when transposed to the realm of real images.

In contrast, Generative Adversarial Networks (GANs) offer a distinct advantage, engendering high-resolution images with fewer artifacts compared to their classical SR counterparts (Wang et al., 2021). GAN-based SR leverages a discriminator network to assess multiple feature values, ensuring that generated images align with the high-resolution standard. Numerous GAN-based SR models have surfaced, employing diverse network components, such as self-attention and transformers (Liang et al., 2021; Zhou et al., 2023), to harness the GAN advantage optimally. Despite their remarkable performance in classical SR, these models collectively underscore the overarching aspiration of SR: the faithful super-resolution of real images (Zhang et al., 2021).

Yet, even amidst the impressive strides made by these models in generating real image SR results, they still grapple with sporadic, unexpected artifacts (Wang et al., 2023; Liang et al., 2022; Xie et al., 2023). These deviations differ from the quandaries of classical SR, where deviations include color discrepancies (see Fig. 5), smeared textures (see Fig. 5), and bent lines (see Fig. 2.) This peculiarity is attributed to the generator's expansive latitude in freely distorting images, a degree of freedom compelling enough to occasionally confound the discriminator (Xie et al., 2023).

Consequently, an intriguing proposition arises: what if the generator's scope for deformation were delimited, allowing it to only subtly alter the input image while outwitting the discriminator within these confines?

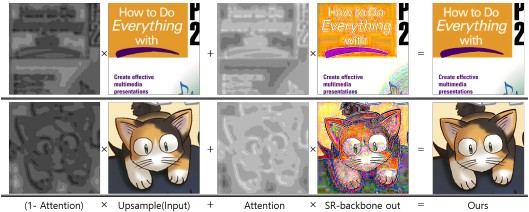

Figure 1: The presented figure showcases illustrative images extracted from key developmental stages of our network. For the purpose of visualization, attention in the central region has been normalized. Best viewed in zoom.

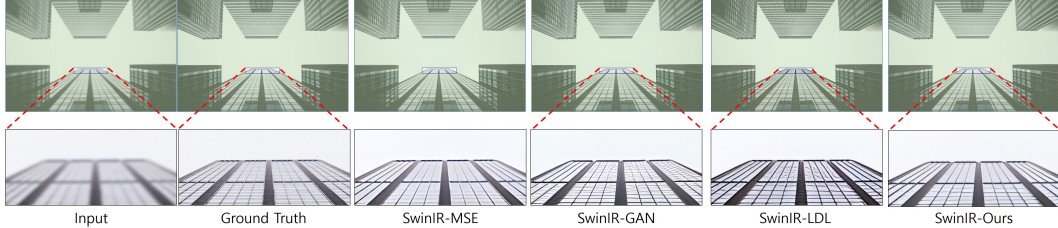

Figure 2: Illustrative example of line distortion in the figure. In the context of the compared networks, we trained the SwinIR-MSE model with the SR backbone. When contrasted with the outcomes of SwinIR-GAN and SwinIR-LDL, both methods significantly induce multiple horizontal distorted lines, whereas our findings distinctly reveal the successful elimination of these line distortions.

This pondering forms the bedrock of our proposal: a strategy to confound the discriminator while minimizing image alterations through the utilization of residual connections and attention maps linked to the original image (see Fig. 1. and Fig. 3.)

Our proposed method's framework offers an avenue to enhance the capabilities of existing super-resolution (SR) models by incorporating them as foundational elements. Our learning process is executed end-to-end, ensuring seamless mapping. To effectively leverage the structural advantages embedded within our approach, precise control over the attention map during the learning process becomes crucial. This necessity is accentuated by integrating pre-trained SR models within the SR backbone, even if their training was rooted in MSE loss. Without appropriate regulation, the attention map tends to converge to a value of 1.0, causing the final SR image to heavily rely on the SR backbone output. To counteract this, we introduce a novel *squeeze and spread losses*, enabling a controlled attention map modulation.

To validate the efficacy of our proposed method, we adopt the validation methodology outlined in Zhang et al. (Zhang et al., 2021)'s blind image super-resolution framework. The low-resolution (LR) images generated through Zhang et al.'s degradation model exhibit notable texture blurring and noise, posing a challenge for traditional SR methods to produce clear SR images. Given this context, we assert that Zhang et al.'s approach is an optimal benchmark for assessing SR images' resilience and distortion profiles, especially when GANs are employed. We performed experiments by implementing our proposed framework with different SR backbones and verified the effectiveness of our approach through the obtained output results.

Our contributions can be encapsulated as follows:

- We present a novel framework for producing super-resolution (SR) images from input images by integrating residual connections and attention maps. The residual connections and attention map support the generated SR image in minimizing distortions as much as possible. To the best of our knowledge, this is the first attempt to use both a residual approach and attention map while generating SR images.

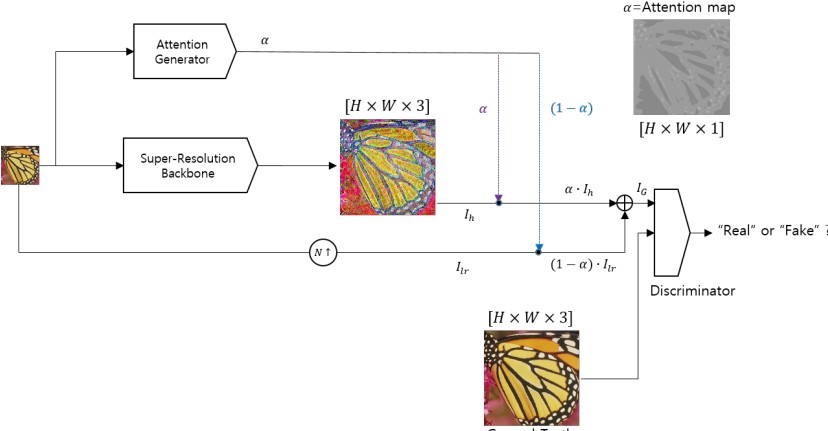

Figure 3: Diagram illustrating the comprehensive structure of our proposed pipeline. The attention map, denoted as $\alpha$, serves as a guiding factor. The residual image, $I_h$, plays a pivotal role, while $I_{lr}$ represents the input image after bilinear upscaling with a scale factor of $N$. The resulting output, $I_G$, represents the super-resolved image attained through our approach.

- We propose a novel architecture for attention generation. We also introduce *squeeze and spread losses*, which substantially enhance attention efficiency.

- Our method effectively corrects distortions in SR results in real images, thereby enhancing the image quality of the SR images.

In essence, our work navigates the convergence of classical SR and GAN-based SR, spotlighting a novel approach that augments perceptual fidelity while preserving image integrity. Through a synergy of network design and loss strategies, we traverse the complexities of real-world image super-resolution, propelling the field toward enhanced outcomes with far-reaching implications.

## 2 RELATED WORKS

### 2.1 CLASSICAL SUPER-RESOLUTION

The classical super-resolution (SR) paradigm involves generating high-resolution images from corresponding low-resolution inputs. This approach commonly employs pixel-value differences, measured by metrics like Peak Signal-to-Noise Ratio (PSNR), to minimize discrepancies between the generated SR image and the ground truth (GT) high-resolution image (Dong et al., 2014; Niu et al., 2020; Sun et al., 2023). Typically, training and evaluation data consist of bicubic down-sampled versions of GT images. The advantages of this approach include the straightforward quantitative comparison with prior results.

### 2.2 REAL-WORLD SUPER-RESOLUTION

Real-World super-resolution focuses on enhancing not only pixel-level accuracy but also the perceptual quality of images. Unlike classical SR, which relies on MSE, this approach employs methods like Generative Adversarial Networks (GANs) or diffusion models (Wang et al., 2021; Zhu et al., 2023; Liang et al., 2021). These techniques address challenges such as restoring intricate textures in low-resolution images, resulting in visually convincing images that resemble real-world scenes (Ledig et al., 2017). However, the generative nature of the network can introduce various distortion problems, including artifacts, blurring, and color shifts.

## 2.3 SUPER-RESOLUTION METRICS

While traditional SR has long relied on metrics like PSNR and Structural Similarity Index (SSIM (Wang et al., 2004)) for evaluation, concerns have arisen about their adequacy in assessing the true restoration quality of SR images (Blau & Michaeli, 2018). An alternative metric, Learned Perceptual Image Patch Similarity (LPIPS (Zhang et al., 2018b)) , measures the deep feature distance between SR images and GT. The LPIPS metric is often preferred when the goal is to measure perceptual similarity rather than pixel-wise intensity differences. It can better handle scenarios where images have different structures, textures, or color shifts while still looking perceptually similar (Zhang et al., 2017b; 2016; 2018b). Nevertheless, no singular metric currently exists that can comprehensively gauge the success of super-resolution techniques (Magid et al., 2022; Jiang et al., 2022).

## 2.4 BLIND-IMAGE SUPER-RESOLUTION

The blind-image SR approach addresses real-world SR challenges beyond network architectures by encompassing a spectrum of potential degradations inherent to real-world scenarios. In contrast to Real-World SR, this method generates low-resolution training images that intricately integrate a diverse array of real-world degradations (Zhang et al., 2018a; 2021; Bell-Kligler et al., 2019). The primary objective is to fortify the robustness and adaptability of SR networks in the face of intricate and unpredictable conditions. In the context of our study, we also introduced the DIV2K4D dataset, as proposed by Zhang et al. (Zhang et al., 2021), and utilized it in our experimental analysis. We examine how the influence of real-world degradations—conditions not explicitly encompassed in the SR network's training—impacts the generated SR images.

## 2.5 DISTORTION ELIMINATION IN GAN BASED SR

In recent years, a growing research interest has focused on mitigating artifacts in Super-Resolution (SR) results using Generative Adversarial Networks (GANs). Liang et al. (2022) introduced an innovative approach involving the creation of an artifact map capable of distinguishing between artifacts arising from GAN-based SR image generation and the fine details of the generated images. The network was then trained to minimize this artifact map, effectively reducing artifacts. Xie et al. (2023), on the other hand, attributed these artifacts primarily to the inherent characteristics of GANs, which tend to overemphasize high-frequency details during the SR process. Xie proposed generating an artifact map to address this issue to ensure that GAN-SR results align closely with MSE-SR results, reducing artifacts caused by excessive high-frequency detail restoration. While Xie's work also acknowledges the challenge of GAN-generated artifacts stemming from an abundance of detail, our approach differs in that we do not employ artifact maps to eliminate GAN-based SR artifacts. Instead, we focus on artifact reduction through the use of attention mechanisms.

# 3 METHODS

Our proposed framework, depicted in the Fig. 3, encompasses several essential components: an attention generator, an SR backbone responsible for generating a residual image, a residual connection section for integrating the original image and the residual image, and a discriminator tasked with comparing the final SR image against the ground truth (GT) image.

## 3.1 SUPER-RESOLUTION BACKBONE

While existing super-resolution (SR) models excel at interpreting and reconstructing images, we recognize that the significant freedom afforded by Generative Adversarial Networks (GANs) often leads to diverse distortions. To mitigate this, we employed the backbone of established SR models to generate residual images without introducing additional structural modifications.

## 3.2 ATTENTION GENERATOR

A comprehensive depiction of this component is presented in Fig. 4. The blue and red blocks encompass convolutional layers (conv), batch normalization (bn), rectified linear units (relu), max-pooling, and 'layer1' and 'layer2' structures, closely mirroring the architectural configuration of

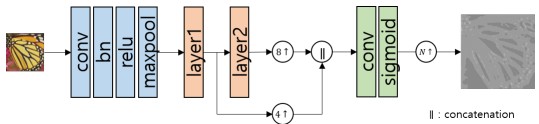

Figure 4: Scheme illustrating the attention map generator within our proposed framework

ResNet50 (He et al., 2016). The feature maps generated by 'layer1' and 'layer2' undergo bilinear interpolation by factors of 4 and 8, respectively, aligning their dimensions with the height and width of the input low-resolution (LR) image. Subsequently, these feature maps are concatenated along the feature dimension. This concatenated feature map undergoes convolution using a 'conv' layer (green block) with a kernel size of $K \in \mathbb{R}^{9 \times 9 \times (256+512) \times 1}$ and padding of 4, where 256 represents the feature dimension of the feature map after 'layer1', and 512 represents the feature dimension of the feature map after 'layer2'. The resulting feature map then undergoes a sigmoid function, culminating in the derivation of an attention map (denoted as $\alpha$) through bilinear upsampling based on the super-resolution (SR) scale factor $N$.

### 3.3 RESIDUAL CONNECTION

The super-resolved (SR) image $I_G$ was generated using three key components: the **attention map** $\alpha$ produced by the attention generator, the residual image $I_h$ generated by the SR backbone, and the bilinearly interpolated version of the input image $I_{lr}$. The resulting SR image, denoted as $I_G$, is defined by the equation:

$$I_G = \alpha \cdot I_h + (1 - \alpha) \cdot I_{lr}, \tag{1}$$

where $\cdot$ is a hadamard product. This generated $I_G$ was subsequently utilized as input for the discriminator network.

### 3.4 DISCRIMINATOR

Within our proposed framework, the discriminator assumes a pivotal role. Its functionality encompasses two critical aspects: guiding the attention map's focus to mitigate input image modification and facilitating the generation of $I_h$ to ensure that the resulting $I_G$ remains indistinguishable from the ground truth image. Through extensive experimentation, we determined that the U-Net architecture aligns most effectively with our approach. As a result, we selected U-Net (Schonfeld et al., 2020) as the discriminator to fulfill these tasks.

### 3.5 OPTIMIZATION

In the framework proposed in Fig. 3, if there are no constraints on the generation of attention maps, the values of all attention maps tend to converge to 1. This phenomenon arises because we employ pre-trained models from the classical SR method as the SR backbone. Despite updating all parameters in an end-to-end manner, the reliance on the initial SR backbone remains significant, as it already produces higher-resolution images than the original inputs. To ensure that the attention map aids in achieving optimal SR performance while minimally altering the input image, we introduced the following "squeeze and spread" losses, denoted as $L_{ss}$:

$$L_{ss} = \gamma \cdot \|\alpha\|_F - \frac{1}{H \times W} \sum_{i=0}^{H-1} \sum_{j=0}^{W-1} \langle \alpha_{i,j} \cdot \log(\alpha_{i,j}) \rangle, \tag{2}$$

In Eq. 2, the first term serves to regularize the attention map $\alpha$, while the second term disperses the values in $\alpha$ to prevent mediocrity. This squeeze and spread loss is then integrated to form the final generator loss:

$$L_G = -\mathbb{E}_z \left[ \log D_{enc}^U (G(z)) + \sum_{i,j} \log \left[ D_{dec}^U (G(z)) \right]_{i,j} \right] \quad (3)$$
$$+ \eta \cdot L_{ss},$$

Here, $\left[ D_{dec}^U (G(z)) \right]_{i,j}$ refers to the discriminator's decision at pixel $(i,j)$, and $\left[ D_{enc}^U \right]$ is the encoder module of the discriminator as defined in (Schonfeld et al., 2020). We use 0.1 for $\gamma$ and 0.01 for $\eta$.

## 4 EXPERIMENTS

### 4.1 IMPLEMENTATION DETAIL

In our proposed framework, all parameters are learned in an end-to-end manner. To ensure a fair comparison, we initialize the parameters of the SR backbone model used for training with those from a pre-trained model designed for classical SR objectives (MSE-SR). This approach aligns with previous studies that have successfully trained Real-World SR models using GANs.

The parameters of the red and blue blocks within the Attention Generator, as shown in Fig. 4, are also initialized using the pre-trained model's parameters. Specifically, these parameters are sourced from the Deep Ten model (Zhang et al., 2017a) that was trained on the minc dataset (Bell et al., 2015). **Note that the overall architecture of attention generator is totally different from the Deep Ten model**.

### 4.2 TRAINING DATASETS

We utilized high-resolution (HR) images sourced from commonly used datasets like DIV2K (Agustsson & Timofte, 2017), Flick2K (Lim et al., 2017; Timofte et al., 2017), and OST (Wang et al., 2018a) to train our super-resolution (SR) models. Instead of using the low-resolution (LR) images provided directly by these datasets, we opted to generate LR images using the degradation model introduced by Zhang et al. (Zhang et al., 2021). This decision stemmed from our experimental findings, which demonstrated that models trained with LR images generated through Zhang et al.'s method yield SR images with enhanced robustness against noise and blur.

### 4.3 TESTING DATASETS

To assess the efficacy of our proposed approach, we carried out comprehensive experiments using two distinct dataset types: DIV2K4D data and RealSRSet (Zhang et al., 2021). Since the DIV2K4D data is not readily accessible in the public domain, we meticulously generated it by applying the procedure outlined in Zhang et al (Zhang et al., 2021). Notably, DIV2K4D comprises high-resolution (HR) image pairs, enabling a quantitative performance evaluation. On the other hand, while Real-SRSet contains pertinent data, it lacks a GT reference, limiting evaluations to a qualitative nature.

### 4.4 COMPARED METHODS

In this experiment, we conducted a comparative analysis involving BSRGAN (Zhang et al., 2021), REAL-ESRGAN (Wang et al., 2021; 2018b), SwinIR (Liang et al., 2021), and our method using SwinIR as the SR backbone for the RealSRSet. The objective was twofold: firstly, to identify any distortions generated during the generation of Real-World SR images by previous methods, and secondly, to assess the efficacy of our proposed method in effectively mitigating these distortions.

Subsequently, we compared our method with Locally Discriminative Learning (LDL) Liang et al. (2022) and vanilla version of the state-of-the-art SR backbones (BSRGAN, SwinIR, and SR-Former (Zhou et al., 2023)) on the DIV2K4D dataset. The primary focus of this experiment was to validate the seamless compatibility of our proposed method with diverse SR backbones and to quantitatively assess the overall enhancement imparted by our proposed method to the image quality.

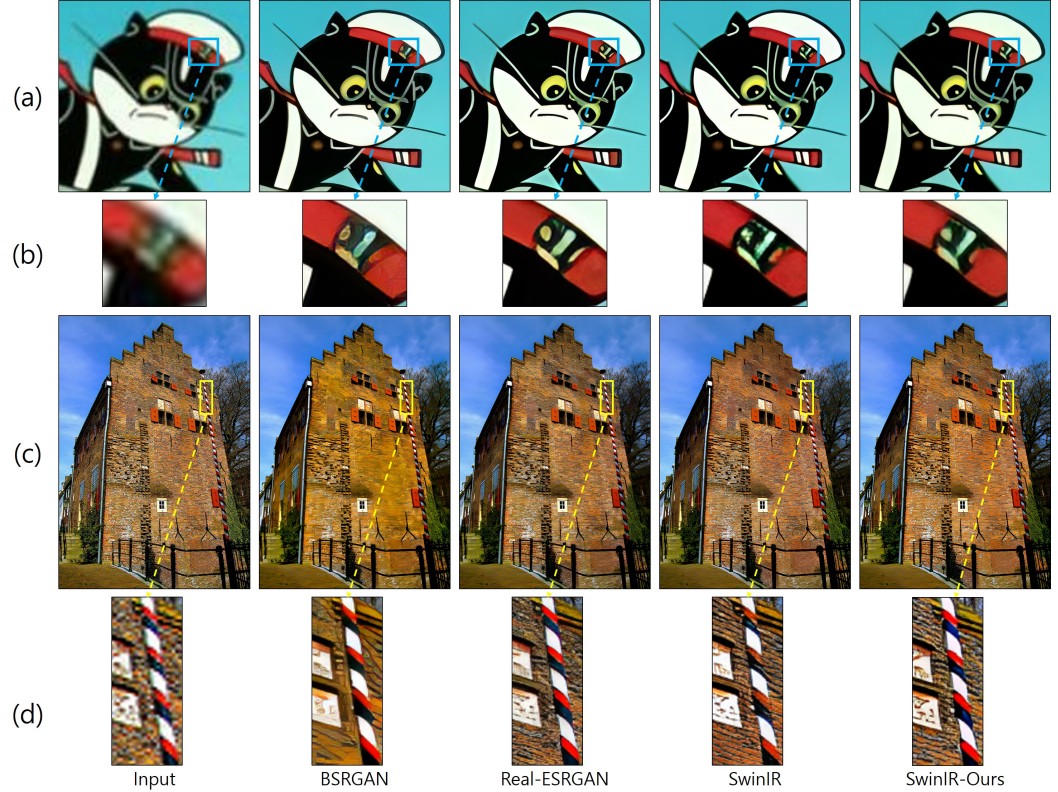

Figure 5: Illustrative example highlighting color distortion and texture smearing. In the second row of Real-ESRGAN (Wang et al., 2021; 2018b) results, the circle on the right, initially red, has undergone a distortion leading to a yellow coloration. Similarly, in the third row with BSRGAN (Zhang et al., 2021) results, a discernible shift in the overall building color is evident. Moving to the fourth row featuring SwinIR, the texture of the surrounding brickwork has blended into the striped pattern of the chimney. We used SwinIR (Liang et al., 2021) as our SR backbone in here. Best viewed in zoom.

## 4.5 EXPERIMENTS ON THE REALSRSET DATASET

We utilized SwinIR as the SR backbone for integrating our proposed method in this specific experiment. Given the absence of Ground Truth (GT) in the RealSRSet, our objective was to directly compare the resultant images to assess the impact of our method on the SR output. Upon close examination, as illustrated in Figures 2 and 5, our proposed approach effectively addresses issues such as bent lines and smeared texture distortions. While false coloration is not a prevalent concern when utilizing SwinIR, it is notable that our method's efficacy in mitigating false coloration for BSRGAN and Real-ESRGAN.

## 4.6 EXPERIMENTS ON THE DIV2K4D DATASET

In this experiment, we aimed to assess the efficacy of our proposed framework across various SR models. Specifically, we compared the performance using PSNR, SSIM, and LPIPS metrics. Given the potential for GAN-generated images to exhibit false coloration (see Fig. 5), we extended our analysis to include PSNR values for RGB channels, as illustrated in Table. 1. The results consistently demonstrate that our proposed method improves PSNR and SSIM values, albeit with a relative weakness in LPIPS values.

Upon scrutinizing the outcomes of multiple SR results, a notable pattern emerged: LPIPS values tend to improve in tandem with increased object contrast and more prevalent high frequency details. This trend suggests that if GAN-based SR models were trained without constraints, as in previous

| | BSRGAN | BSRGAN-LDL | BSRGAN-OURS | SWINIR | SWINIR-LDL | SWINIR-OURS | SRFORMER | SRFORMER-LDL | SRFORMER-OURS |
|---|---|---|---|---|---|---|---|---|---|
| | | | | | PART1 | | | | |
| PSNR | 24.42 | 24.31 | 25.05 | 24.88 | 24.32 | **25.69** | 23.98 | 23.89 | 24.14 |
| SSIM | 0.6969 | 0.6977 | 0.7061 | 0.7142 | 0.7183 | **0.7211** | 0.6606 | 0.6683 | 0.6734 |
| LPIPS | 0.2599 | 0.2497 | 0.2846 | **0.2221** | 0.2225 | 0.2394 | 0.275 | 0.2661 | 0.2669 |
| | | | | | PART2 | | | | |
| PSNR | 24.16 | 24.01 | 24.53 | 23.91 | 23.11 | **24.94** | 23.67 | 22.98 | 23.85 |
| SSIM | 0.6622 | 0.6618 | 0.6637 | 0.6602 | 0.6564 | **0.6759** | 0.6263 | 0.6199 | 0.6375 |
| LPIPS | 0.3023 | 0.2923 | 0.317 | **0.2719** | 0.2766 | 0.2888 | 0.3148 | 0.3099 | 0.3144 |
| | | | | | PART3 | | | | |
| PSNR | 23.48 | 23.29 | 24.35 | 23.55 | 23.02 | **24.47** | 23.34 | 22.96 | 23.49 |
| SSIM | 0.6383 | 0.6356 | 0.6572 | 0.6437 | 0.6488 | **0.6584** | 0.6187 | 0.6201 | 0.6288 |
| LPIPS | 0.3059 | 0.2959 | 0.3204 | 0.2817 | **0.2792** | 0.292 | 0.3133 | 0.3097 | 0.3061 |
| | | | | | PART4 | | | | |
| PSNR | 21 | 21.01 | 21.53 | 21 | 20.73 | **21.89** | 21.16 | 20.98 | 21.45 |
| SSIM | 0.537 | 0.5296 | 0.5501 | 0.5481 | 0.5531 | **0.5677** | 0.5364 | 0.5299 | 0.5433 |
| LPIPS | 0.4394 | 0.4367 | 0.4563 | **0.4092** | 0.4363 | 0.428 | 0.4401 | 0.4399 | 0.4433 |

Table 1: Comparative Evaluation of PSNR, SSIM, and LPIPS Scores across Different Sections of DIV2K4D (Zhang et al., 2021) using State-of-the-Art SR Backbones. The highest-performing SR outputs for each section are highlighted in **bold**. **Note that we utilized PSNR-RGB instead of PSNR-Y, as false coloration in SR output is a significant distortion in GAN-based SR.**

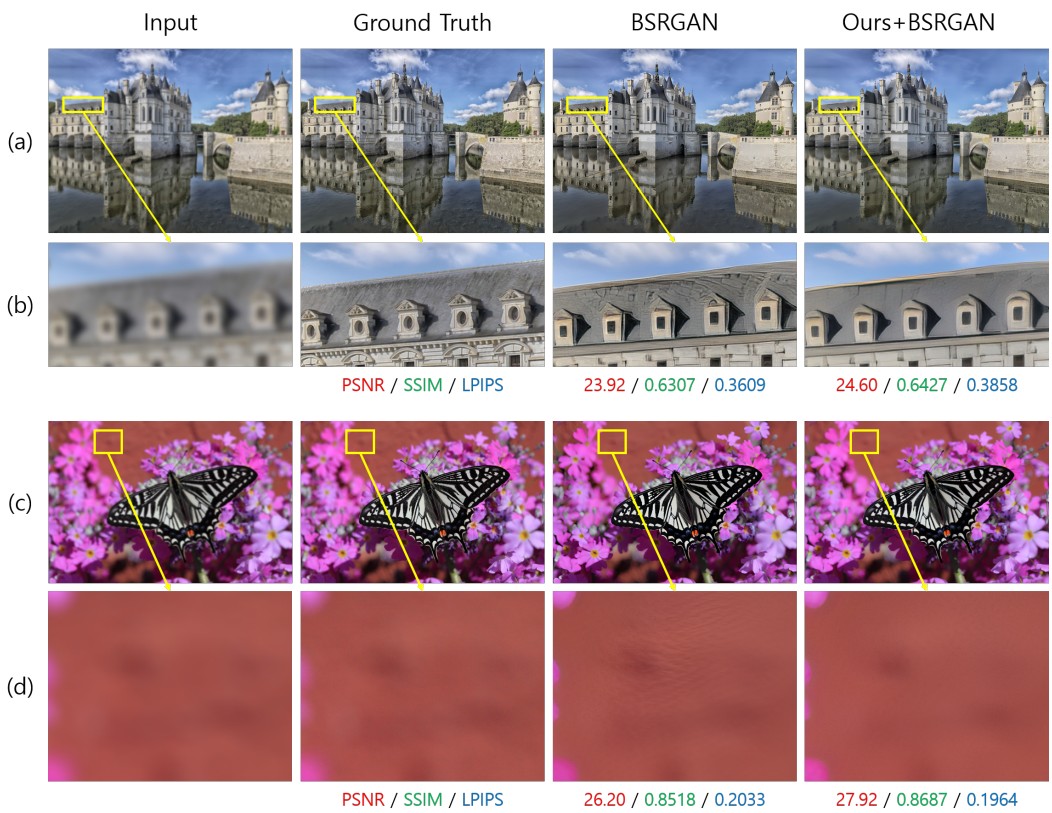

Figure 6: This visual representation illustrates the interaction between local distortion and the LPIPS metric. Notably, the second and fourth lines of the figure reveal that the utilization of BSRGAN introduces noticeable instances of local distortion within the image. Upon closer examination of the results produced by BSRGAN in the fourth row, a distinct wavy pattern becomes apparent in the image. Interestingly, despite the evident presence of these pronounced local distortions, the LPIPS metric does not assign them considerable importance within its evaluation framework. In this depiction, we employed BSRGAN as our chosen SR backbone. For optimal viewing, zooming in is recommended.

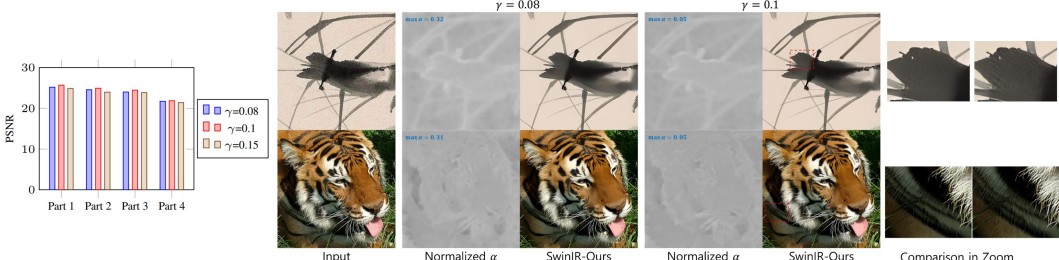

Figure 7: Examples of our results obtained with different gamma values. In our proposed model, a $\gamma$=1.0 yields the highest average PSNR value for the DIV2K4D dataset (left). While a $\gamma$=0.08 implies reduced constraints (higher attention values) during training, the model trained with a $\gamma$=0.1 excels at representing high-frequency details (right).

studies, the resulting SR images might yield even better LPIPS scores. However, Fig. 6 underscores a critical concern—the prevalence of local distortion in previous GAN-based SR model outputs. Paradoxically, these distortions exert positive influence on the computed LPIPS value. Thus, while LPIPS is a valuable perceptual quality metric, its reliability is not absolute.

### 4.7 ABLATION STUDY

We also investigated the impact of the $\gamma$ in our model on the resulting SR image. Referring to Eq. 2, we initially anticipated that a lower $\gamma$ would grant more freedom to the generative aspect of GAN-SR, potentially enabling enhanced high-frequency details. However, as depicted in Fig. 7, the SR image generated with a $\gamma$ of 1.0 exhibits more intricate high-frequency details than the one produced with a $\gamma$ 0.8. Furthermore, the average PSNR for a $\gamma$ of 1.0 surpasses that of a $\gamma$ of 1.5. This shows the significance of a proper balance between the first term, which aims to reduce the overall attention value, and the second term, which prevents a uniform distribution of attention values throughout the process of squeeze and spread losses during the training of our proposed model.

## 5 CONCLUSION

This paper introduces a fresh approach to dealing with the distortions commonly found in realistic super-resolution (SR) images. Our method is about training a GAN to retain the essence of the original image while tricking the discriminator. We have leveraged the original image's residual connections, attention maps, and proposed squeeze and spread losses. These losses help guide the attention generator to operate effectively. A key feature of our method is its versatility across various SR models. Through thorough experiments, we consistently showed how our method can fix many distortions often seen in SR images. This leads to a significant boost in overall image quality. By combining these innovative aspects, our proposed approach opens up new possibilities for better performance in diverse SR applications. As the world of super-resolution progresses, our method stands out as a valuable addition, making it possible to create higher-quality SR images while staying true to the original content.

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

# A    APPENDIX

You may include other additional sections here.

