# OpenReview forum: "Reducing distortions in Real World Image Super Resolution using Attention"
_ICLR.cc/2024/Conference — ICLR 2024 Conference Withdrawn Submission_

### Official Review · Reviewer_X3jk · 2023-10-27

**Soundness:** 3 good
**Presentation:** 3 good
**Contribution:** 2 fair
**Rating:** 3
**Confidence:** 4

**Summary:**

This manuscript improves the performance of the current SR method and reduces the distortion caused by GAN through a mask attention module. Experimental results demonstrate the effectiveness of the proposed method. However, there are still some significant weaknesses.

**Strengths:**

This manuscript proposes a simple method can be used to improve perceptual quality while minimizing distortions in real-world SR images by using existing SR models as backbone.

**Weaknesses:**

The main contribution is a mask channel which is added to a SR backbone. The learned mask can adjust the weights of original content and generated contents. Hence, it can take a balance between original image and generated results.
By comparing the results, we can find that the SWINIR with proposed mask obtains higher PSNR/SSIM but lower LPIPS. However, we can obtain similar results by mixing L1 and adversarial losses or assembling pixel-wise optimized model and GAN-based model. Moreover, from visual comparison in Fig.5 and Fig.6, it can be found that the visual quality of the proposed method is not very satisfactory.
What we expect is that this method can not only reduce distortion (better PSNR/SSIM), but also improve subjective quality and get better LPIPS. Unfortunately, current experimental results can not verify this point.

**Questions:**

In my opinion,  the novelty and insight of the paper is limited to some extent. Hence, the authors can further elaborate on the contribution and innovation of this manuscript.

---

### Official Review · Reviewer_Vfas · 2023-10-29

**Soundness:** 4 excellent
**Presentation:** 3 good
**Contribution:** 1 poor
**Rating:** 3
**Confidence:** 4

**Summary:**

In this paper, the authors utilize residual connections and attention maps to reduce artifacts and distortions associated with previous GAN-based models. The various experiments were conducted to demonstrate the superiority of the proposed methodology.

**Strengths:**

- The method is simple but efficient for PSNR and SSIM on the Table 1.
- The Figure 7 delivers good analysis for the attention map of SR model.

**Weaknesses:**

- Reducing distortion does not necessarily guarantee a human perceptual super resolution result. [1] There is a trade-off relationship between the two, and the good solution is to push this trade-off line. However, according to the results in Table 1, this methodology appears to be sacrificing the LPIPS score in order to increase PSNR and SSIM.
- Explicitly utilizing attention maps is a method that has already been sufficiently attempted in other tasks. [2] The reason why such a methodology has not been introduced in the super resolution in this manner is that it is not the sufficient for adequately solving the above problem.
- There are too limited qualitative results.

**References**
- [1] Blau, Y., & Michaeli, T. (2018). The perception-distortion tradeoff. In Proceedings of the IEEE conference on computer vision and pattern recognition (pp. 6228-6237).
- [2] Alami Mejjati, Y., Richardt, C., Tompkin, J., Cosker, D., & Kim, K. I. (2018). Unsupervised attention-guided image-to-image translation. Advances in neural information processing systems, 31.

**Questions:**

- What is $F$ in the equation 2?
- Is the model trained only with the pixel gan loss and squeeze and spread loss without MSE loss?
- This method seems like it could be applied to not only real-world super resolution but also general single image super resolution. Why then, are there no experiments conducted on DIV 2K validation or similar datasets?

---

### Official Review · Reviewer_NLs5 · 2023-10-31

**Soundness:** 2 fair
**Presentation:** 1 poor
**Contribution:** 2 fair
**Rating:** 3
**Confidence:** 5

**Summary:**

This study addresses a significant issue: real-SR results often contain various artifacts. The authors propose using a residual map to reduce these artifacts, presenting a strategy that can be generally applied during the training stage. They support their approach with compelling observations, such as the example in Figure 2, which underscores the motivation behind their work. This method not only mitigates the artifacts in real-SR results but also enhances the overall quality of the super-resolved images.

**Strengths:**

1. It is reasonable and interesting to employ prior information to guide the generator by an attention generator.

2. As demonstrated in the paper, the attention generator can produce a sensible attention map.

**Weaknesses:**

1.	In section 3.3, it seems like the operation of adding the LR may introduce the degradation of input to the SR results. If this operation introduces degradation, it would be helpful to explain why this happens and how it affects the results.
2.	In section 3.5, It would be beneficial to mention any related work or studies that have used a similar loss function. In addition, adding a discussion will give your work more credibility.
3.	In Tab. 1, If the SR model with the LDL version performs worse than the corresponding SR model, an explanation should be provided.
4.	In Tab. 1, it seems like your proposed method (e.g., the performance on BSRGAN) is a trade-off operation, improving the PSNR performance while sacrificing the LPIPS performance. It would be helpful to discuss why this trade-off occurs.
5.	Consider revising your paper for clarity and correctness. Addressing any spelling errors (e.g., SWINIR->SwinIR) will make your paper easier to read.

**Questions:**

See the above weakness.

---

### Official Review · Reviewer_JhX2 · 2023-11-01

**Soundness:** 2 fair
**Presentation:** 3 good
**Contribution:** 2 fair
**Rating:** 5
**Confidence:** 5

**Summary:**

The authors improve the perceptual quality by strategically using residual connections and an attention map in real-world SR images. It is a simple and versatile framework. The authors design an architecture of attention generator and introduce squeeze and spread losses. The proposed method seems can correct distortions in SR results in real images.

**Strengths:**

The authors propose a simple but versatile framework. It seems that this framework can be generalized to other baselines or other SR methods and drive performance improvements. The aim to enhance the perceptual quality is meaningful. The experimental results that reported by the authors seems have good improvements.

**Weaknesses:**

Although the motivations of proposed method seem make sense, there are still some doubts and concerns remain:
1. Why does adding the simple residual connections and the attention map lead to performance improvements? The author should try to explain, or show it at the level of the feature map.
2. From the paper, LPIPS seems not reliable enough, but the authors should give more evidence to show that, after all, the advantages of the authors' results are not visible from the figure. In Figure 6, the results produced by the author seem to be smoother, but I can't make sure which result is better compared with the other methods.
3. It seems the authors ignore to provide details about the training strategies.
4. The author's ablation experiments are insufficient. The effectiveness of the proposed method cannot be comprehensively proved, such as the effectiveness of the residual connection and attention map, squeeze and spread losses.
5. The layout of some figures and the table in this paper is not standard enough.

**Questions:**

The authors need to solve the questions mentioned in the weakness part.